# Pierre Robin Sequence and 3D Printed Personalized Composite Appliances in Interdisciplinary Approach

**DOI:** 10.3390/polym14183858

**Published:** 2022-09-15

**Authors:** Andrej Thurzo, Barbora Šufliarsky, Wanda Urbanová, Martin Čverha, Martin Strunga, Ivan Varga

**Affiliations:** 1Department of Stomatology and Maxillofacial Surgery, Faculty of Medicine, Comenius University in Bratislava, 81250 Bratislava, Slovakia; 2Department of Oral and Maxillofacial Surgery, Faculty of Medicine, Comenius University in Bratislava and University Hospital, 81372 Bratislava, Slovakia; 3Department of Orthodontics and Cleft Anomalies, Faculty Hospital Kralovske Vinohrady, Dental Clinic 3rd Medical Faculty Charles University, 10034 Prague, Czech Republic; 4Clinic of Pediatric Otorhinolaryngology of the Medical Faculty Comenius University in Bratislava, 83340 Bratislava, Slovakia; 5Department of Histology and Embryology, Faculty of Medicine, Comenius University in Bratislava, 81372 Bratislava, Slovakia

**Keywords:** biocompatible photopolymer resin, 3D printed orthopedic appliance, orthodontics, Tübingen palatal plate, Infants, craniofacial anomalies, micrognathia, airway obstruction, CT

## Abstract

This paper introduces a complex novel concept and methodology for the creation of personalized biomedical appliances 3D-printed from certified biocompatible photopolymer resin Dental LT Clear (V2). The explained workflow includes intraoral and CT scanning, patient virtualization, digital appliance design, additive manufacturing, and clinical application with evaluation of the appliance intended for patients with cranio-facial syndromes. The presented concept defines virtual 3D fusion of intraoral optical scan and segmented CT as sufficient and accurate data defining the 3D surface of the face, intraoral and airway morphology necessary for the 3D design of complex personalized intraoral and extraoral parts of the orthopedic appliance. A central aspect of the concept is a feasible utilization of composite resin for biomedical prototyping of the sequence of marginally different appliances necessary to keep the pace with the patient rapid growth. Affordability, noninvasiveness, and practicality of the appliance update process shall be highlighted. The methodology is demonstrated on a particular case of two-year-old infant with Pierre Robin sequence. Materialization by additive manufacturing of this photopolymer provides a highly durable and resistant-to-fracture two-part appliance similar to a Tübingen palatal plate, for example. The paper concludes with the viability of the described method and material upon interdisciplinary clinical evaluation of experts from departments of orthodontics and cleft anomalies, pediatric pneumology and phthisiology, and pediatric otorhinolaryngology.

## 1. Introduction

Research into advanced biocompatible polymers and advanced manufacturing has brought new manufacturing approaches. Today, advanced manufacturing based on biocompatible polymers and composites is moving from research to clinical practice. Clinical implementations of these materials help to improve the lives not only for those needing prosthetics or patients suffering tissue loss or disfigurement, but also enable early treatment in children with cranio-facial syndromes, including micrognathia in Pierre Robin sequence (PRS) [1,2].

Micrognathia is a condition in which the lower jaw is undersized. It is a symptom of a variety of craniofacial conditions, not only PRS. Sometimes called mandibular hypoplasia, micrognathia may interfere with child’s feeding and breathing (Figure 1). Specific situations in growing children with rapidly changing morphology require personalized biocompatible appliances respective for sequences of adapted variants to keep the pace with their growth. Frequently, situations in patients with cranio-facial syndromes, including Pierre Robin sequence (PRS), result in breathing problems. If the airway obstruction becomes severe, the situation is life threatening. With proper help, the majority of infants with PRS outgrow breathing problems within three to six months as their airway grows.

The mechanical properties and biocompatibility of current photopolymer resins are advancing every day. Many of them are already excellent for various biomedical applications and are widely used in interdisciplinary cooperations between medical and technical fields addressing different problems [1,2,3,4]. The crucial part of this paper is focused on the clinical application of a personalized appliance 3D-printed from Class IIa long-term biocompatible resin. The material properties are essential for appliance performance and clinical applicability, including patient safety. Without the proper material, any appliance resulting from the workflow described in this paper would be worthless. Historically, the Dental LT Clear material was introduced by Formlabs (Somerville, MA, USA) in April 2017, specifically designed to work with Form 2 printer. This paper describes the utilization of the 2nd generation of this material. The paper’s focus is wider than the sole application of the designed appliance to PRS. The concept provides a workflow applicable to wide range of complications resulting from many different craniofacial syndromes.

Development of the skull can be divided into neurocranial and viscerocranial formation. Fetal head has origin in mesenchymal cells, which originate from unsegmented paraxial mesoderm. The cranial neural crest is formed through a process called epithelial–mesenchymal transformation (EMT). Epithelial cells become mesenchymal in the process [5].

Neurocranial growth via membranous ossification by formation of mesenchymal cells via EMT occurs during the first 8 weeks of pregnancy. Mesenchymal cell condensation allows the development of cranial bones [6].

The whole process is complex and complicated in combination of different progressions depending on the three-dimensional morphogenetic development. Coordination and control are determined in hundreds of different genes, as the tissues of the skull have different embryonical origins. More than one-third of congenital defects are craniofacial abnormalities and malformations [7]. These malformations have multistage impact on the patient and are debilitating, with negative qualitative and quantitative effects on the patient.

Most of the craniofacial disorders (CFD) are morphologic. The neurocranium and viscerocranium have distinct functions, such as protecting the brain and the sensory organs, shaping the human face, and protecting neurovascular structures. Craniofacial disorders therefore affect the shape and function of the head, face, occlusion, airways, aesthetics. The majority of CFDs are associated with other forms of congenital malformation. Etiologically, CFD can be divided into developmental anomalies (such as orofacial clefts, craniosynostosis), affecting speech, airways, development of the brain, senses, and even have social impacts on the patient [8,9,10]. Together with developmental anomalies are chromosomal abnormalities (e.g., Williams syndrome), single-gene disorders, malformation sequences (e.g., Pierre Robin Syndrome) or epigenetic mutations and external mechanical deformations [11].

Multidisciplinary cooperation is required in CFD; the methodology described in this paper points toward the role and need for cooperation among experts from medical and technological fields in the whole process. From the clinical side, it includes departments of orthodontics and cleft anomalies, dentistry, neonatal and pediatric pneumology, pediatric otolaryngology, speech therapy, genetics specialist and maxillofacial surgery, and full compliance of the caregivers. Variations and different stages of severity in CFD need an individual approach, personalization, creativity and a team familiar with up-to-date techniques, implementing modern methods, such as 3D printing, modeling, scanning and simulations, for example, a CAD/NAM printed device for newborns with lip and palate clefts [12]. Biocompatible composites now have a wide spectrum of medical applications including regenerative medicine utilizations of composite scaffolds [13], tissue engineering [14,15], or patient-specific orthoses and craniomaxillofacial implants [16,17]. New biocompatible composites regularly appear on the market, albeit there is a lack of information regarding the mechanical or thermal properties of the printed products. Many of these can be suitable for 3D-printed biomedical devices such as orthoses, casts, medical models, and surgical guides [18,19].

Pierre Robin sequence (PRS) is specific for combination of micrognathia, with related glossoptosis, posteriorly positioned tongue, airways obstruction, and cleft palate with variations in severity. Pierre Robin sequence can be one of the features for different syndromes, such as Stickler syndrome or isolated Pierre Robin syndrome [20]. Each symptom and abnormality results in another. Around 40% of PRS are isolated and around 60% of the patients are addressing other additional symptoms and features of different syndromes. PRS has frequency of 1/8500–14,000 births. With its ratio, PRS is one of the most common birth defects [21].

Early intervention is mandatory as the PRS triad (micrognathia, glossoptosis and airway obstruction) brings wide scale of additional complications, including high risk of death or brain damage in the infant. Cleft palate varying in its severity requires either surgical treatment or conservative treatment by obturator followed by surgical treatment. A cleft palate, depending on severity, brings complications such as ear infections, food inhalation and airway infections, and can result in speech problems too in older children. Micrognathia has no specific definition in newborns and infants; it can only be evaluated and confirmed by the doctor, specifically by the doctor’s professional opinion. Micrognathia in combination with physiological tongue size, or together with larger tongue, with some severity of glossoptosis, cause problems not only with feeding but most importantly with breathing and airway patency. Airway obstruction in this case is most probably caused by combination of glossoptosis and lack of space for suprahyoid muscles to pull the airways, in particular geniohyoid muscle attached to hyoid bone, which pulls lower airways upward and forward helping to widen airways. With micrognathia, the position of suprahyoid muscles, especially geniohyoid muscle originating on the inferior mental spine of the mandible and inserted into hyoid bone, is insufficient. In newborns and infants, as well as others, the upper airway obstruction causes a serious risk of sudden death syndrome and serious brain damage, also producing a negative impact on cognitive development from a lack of oxygen [22].

The diagnosis of the airway obstructive sleep apnea (OSA) must be determined by polysomnography considering the anatomical cause, also to differentiate from a central cause, or to diagnose mixed apnea. Effectiveness of the treatment is evaluated by polysomnography after correct diagnosis and correct choice of treatment. After the cause of the apnea is determined by polysomnography, infants must undergo examination at otorhinolaryngology. The airways, airway obstruction, its severity, extent of the cleft palate, and tongue position and its relation to the obstruction are examined during fiberoptic nasolaryngoscopy with no sedation, or in drug-induced sleep endoscopy. Full visualization of the upper airways with the possibility of video recording and possibility to measure anomalies gives an opportunity to evaluate components of OSA at the level of the nasal cavity, pharynx, and larynx.

Obstructive sleep apnea treatment varies from non-invasive treatment to different variations of surgical treatment. Surgical treatment requires general anesthesia, which brings other potential risks. Articles describing the negative effect of general anesthesia are contradictory, as most clinical studies are retrospective. Some studies have shown that general anesthesia can cause neurocognitive deficits in children; some have not, but generally practitioners do not recommend that newborns undergo general anesthesia at a very early age to avoid the risk of such a negative impact on neurocognitive development [23].

Variations of the treatment are possible. In the decision for diagnosis and treatment, different factors are considered, such as sleep quality and apnea, breathing, the possibility and compliance with the feeding of the infant, growth, physical and psychomotor development. Non-invasive treatments such as nasopharyngeal tube, application of positive airway pressure (CPAP) and application of Tübingen palatal plate bring the desired results and effectiveness. Tübingen palatal plates have been proved to be effective in the treatment of OSA without invasive surgical procedures [24].

Surgical treatment of obstructive sleep apnea caused by glossoptosis and mandibular hypoplasia is accomplished through mandibular distraction osteogenesis, tracheostomy, attaching the tongue to the lower lip, and tongue–lip adhesion [20]. All surgical procedures bring various risks and complications, such as infection of the surgical wound, or fibrous nonunion in the case of mandibular distraction osteogenesis. In the case of tracheostomy, there is a risk of accidental decannulation causing death, tracheal stenosis or persistent percutaneous tracheal fistula, requiring reconstruction [25].

The concept of the Tübingen palatal plate (TPP) as a removable appliance brings an effective and non-invasive way for the treatment to solve more than one problem for infants with PRS. Alleviating upper airways obstruction is crucial for intellectual development. Through a reduction in respiratory complications by closing the cleft palate and by pressure applied to the root of the tongue resulting in increased tongue-pressure on the frontal-lingual area of the mandible, TPP might induce mandibular growth [24]. Standard procedure for treatment with TPP starts with impressions of the upper jaw; afterwards, a prototype of TPP is created in a dental lab by a technician in form of a palatal base covering the cleft, alveolar ridges and including extraoral extension for fixation and extension, the spur, with individual length, which is standardly around 3 cm long and has a function to push the root of the tongue forwards. TPP is inserted under the supervision of the video-fibro-endoscopy [24].

If there is a necessity for modifications such as changes in length, angle or stability, the TPP must be revised until the perfect fit and desired effect is achieved. The patient must undergo video fibro-endoscopic recalls and repeated hospitalization until the TPP has the proper size, shape, and length of the spur and the desired position on the tongue and airways is reached. Three-dimensional intraoral and extraoral scanning and 3D printing bring opportunities to lower the risks of human error, lower the time for preparation of the TPP, even possibly predicting growth by setting up the offset when modeling and printing. Infants grow rapidly, especially during the first year of life. After each short period of life, the appliance prototype becomes inapplicable.

Intraoral scanning is a fast, safe, and feasible procedure for neonates, small children, and infants with craniofacial malformations and is critical practically for all 3D designed and printed individual appliances [26,27,28]. Infants from birth have many growth spurts, especially during the first year of life. After each spurt period of the infant, the prototype becomes inapplicable.

Certified biocompatible resins for 3D printing, suitable for long-term use in the oral cavity, are on the market and are easily accessible. The possibility of use of 3D printing with biocompatible resin has the advantage to rapidly create series of devices at once. The properties of the material, such as rigidity, flexibility, and durability, can improve the patient´s comfort, the desired effect on the tissues and reduce the need for extra safety features.

The aim of this paper is to introduce this approach to clinicians. To achieve this objective, the method and concept are explained in:a clinical interdisciplinary contexta biomedical context for the practical appliance designpractical additive manufacturing with post-processing and application method

The goal is to approximate clinicians to the described technologies and explain availability of the described workflow, which directly allows them to design and 3D print personalized appliances made from the investigated resin without the need of a specialized dental laboratory.

Sharing this knowledge with other clinicians will improve the outlook for patients with craniofacial syndromes. These patients frequently face life-threatening situations in their infancy or suffer significant brain damage from hypoxia resulting from airway obstruction caused by micrognathia-related glossoptosis. The current typical medical intervention to these infants is a risky permanent surgical tracheostomy. As an alternative option, the TPP appliance designed by lab technician does not easily keep up with the rapid growth of the infant. The aim of this paper was to introduce a novel concept that utilizes CT and intraoral scan. After segmentations of the CT that provide 3D morphology of the face, lips, intraoral cavity, and oropharynx, the 3D outputs merged with intraoral scan allowed the design of an orthopedic appliance. The 3D printing of such an appliance is not new [27,29,30,31]; however, the design of extraoral part of the appliance and TPP spur upon segmented CT is a new approach with clear benefits to the patient. The aim was to provide a workflow that would allow us to update the alveolar part of the appliance frequently as the infant grows. The goal was to achieve this without extra invasiveness and while preserving the affordability and practicality of the workflow. The design of the appliance is not the key aspect of this paper. With accurate knowledge of anatomical parts described above, any appliance can be designed. The key part of this paper was finding a reasonably resilient material that would be certified, dedicated for 3D printing, biocompatible for permanent intraoral application, and at the same time be durable and resistant to fracture.

The novelty that should be emphasized is that the full design and in-house 3D printing of a custom orthopedic device from Dental LT Clear (V2) biocompatible photopolymer resin can be used to fabricate the entire device.

While appliance customization and personalization costs are higher than mass-produced universal devices, the user-friendly workflow for in-office production of these medical devices has proven to have the potential to improve interfaces for non-invasive ventilation for neonates and small infants, as well as increased their comfort [32].

The workflow does allow direct 3D design of appliances by doctor and direct in-office biocompatible printing without the necessity of a lab technician intermediary. The method affordably provides a sequence of variant appliances intended for change every 14 days upon renewed intraoral optical scan without changing the extraoral part or intra-pharyngeal spur design. The applicability of this fully evaluated photopolymer for removable appliances intended for permanent intra- and extraoral contact is a valuable message to clinical practice.

## 2. Method and Material

The modus operandi presented in this paper includes a description of interdisciplinary cooperation from various medical fields that leads to the final appliance design, manufacturing, and clinical application with evaluation. The complex scheme of processes covered by this paper is shown in Figure 2.

The paper does not focus on explanation of common CT segmentation or simple alignment of the intraoral scan with segmented CT data. Biocompatible composite material properties are crucial for the appliance to achieve the desired clinical effect. In the case of PRS, it is crucial for the appliance to be able to propel the mandible and root of tongue from the collapsed pharyngeal part of the upper respiratory tract. The most relevant properties of the 3D-printed appliance for similar clinical applications are:Durability with reasonable hardness (>75D)Resistance to fracture (Total fracture work ≥ 250 J/m^2^)Flexural Strength (at 5% strain ≥ 80 MPa)Flexural Modulus (≥2000 MPa)At least Class IIa biocompatibilityNo cytotoxicity or genotoxicity, not an irritant or a sensitizer

### 2.1. Clinical Aspects: Interdisciplinary Cooperation and Principles of the Concept and Method

The concept of this paper emerges from the clinical problems of rapid growth and minimal compliance of infants with PRS and the urgency to effectively address frequently severe, life-threatening symptoms such as airway obstruction. Methodology and interdisciplinary cooperation are explained in a real case of PRS—an infant with micrognathia, glossoptosis and airway obstruction. The methodology section first describes the interdisciplinary approaches of experts from departments of orthodontics and cleft anomalies, pediatric pneumology and phthisiology as well as pediatric otorhinolaryngology. Scanning of intraoral structures was performed with an iTero^®^ Orthodontic Scanner (Align Technology, San Jose, CA, USA), the face surface was segmented from the full head CT scan with a resolution of 0.3 mm and scans were processed in Meshmixer™ (Autodesk^®^, Inc., San Rafael, CA, USA). Examples of the creation of two-part biocompatible 3D-printed appliances based on the merged intraoral and extraoral soft-tissue scan are presented. The clinical feedback evaluation of the presented clinical example suggests the need for an update of the intraoral appliance every 14 days in a two-year-old infant to preserve comfort and thus infant compliance with the appliance.

Interdisciplinary cooperation and the management of such patients is different in each country. Whether cleft palate surgical therapy involves maxillofacial surgery or a plastic surgeon depends on the country, although in such cases including micrognathia and glossoptosis, the complexity of the case is likely to necessitate the maxillofacial surgery department. The certainty is that perfect cooperation and work coordination between different fields is required. Pediatric pneumology and phthisiology departments are required for diagnostics of airway obstruction, differentiating the type of sleep apnea and polysomnography, and the pediatric otorhinolaryngology department for diagnostics of localization of upper airway obstruction. Experts from the department of orthodontics and cleft anomalies cooperate with other specialists from dentistry to diagnose the severity of the disability and evaluate the viability of appliance therapy. Speech therapists are also helpful in infants supporting acquisition of the sucking habit, and naturally in older cases such as that presented. Specialists in genetics help to diagnose other additional syndromes. The chief complaint must be diagnosed and described by specialists in each medical field included in the team before taking any further steps in therapy and treatment.

An exemplary case of a patient with PRS was used for demonstration of interdisciplinary cooperation, appliance design, manufacturing and clinical application followed by evaluation. The patient was two years old and already hospitalized. It was also the first patient for whom the TPP was created from the very beginning to avoid tracheostomy. Each case must be individually considered and in this exemplary case, the need for a precise appliance was considerable without any extra time for mistakes or trials. To be accurate, to have the ability to create a spur of the exact needed length and shape, to diagnose and to produce 3D analysis of the skull and airways, CT imagining was crucial. The level of invasiveness of the TPP preparation, after use of CT, was higher than the classical process of technician-created TPP, albeit the appliance’s anatomical complementarity and accuracy with the extraordinary speed of creation revealed the advantages of this method. The desired effect on the airways is typically reached by use of the first set of TPP.

The toddler underwent examinations at different clinics prior to the evaluation for suspicion of sleep apnea at the Department of pediatric pneumology and phthisiology together with the Clinic of pediatric otorhinolaryngology.

The whole scale of craniofacial disorders, especially ones affecting the viscerocranium, must be examined for possible obstruction of the airways, to avoid the risk of brain damage, neurocognitive development delay, sudden death, or malnutrition. Untreated sleep apnea and airway obstruction leads to irreparable changes in the brain.

The severity and range of the airway obstruction in the toddler was examined and determined by a polysomnography examination. Polysomnography is lege artis and the gold standard for confirmation of diagnosis of airways obstruction and the type of sleep apnea, whether it is obstructive sleep apnea, central apnea, or mixed apnea. The device used in the case was from Löwenstein Medical Technology GmbH & Co, Hamburg.

Polysomnography as a systematic procedure includes electroencephalography, electrocardiography, pulse oximetry, evaluation of saturation (SpO2) and pulse, airflow and respiratory patterns, control of the chest and abdominal wall movements, nasal pressure and airflow, and videorecording. The procedure lasts 8 h or more as standard.

The evaluation of the toddler also involved a pediatric otolaryngology specialist, to evaluate and diagnose deformity and malformations of the nasal cavity and palate, severity of the cleft, pharynx, and larynx anatomical and functional conditions, including the presence of malformations and swallowing disorder. An ENT (ear, nose, throat, and neck) examination was executed on the unseated patient by fiberoptic video nasolaryngoscopy. The position of the tongue, size of the root and the body of the tongue was assessed, as well as its movements, its relationship to the palate, swallowing and breathing, range of the cleft palate and severity of cleft affecting the nasal cavity and the space between the root of the tongue and hypopharynx.

The child had already been treated with continuous positive airway pressure (CPAP) but she was not tolerating therapy since she was removing the apparatus from her face and there had also been no significant effect on the apnea. After determining the type of the apnea and obtaining positive results for obstructive sleep apnea, the treatment by TPP was chosen as the primary treatment for airway obstruction. Specialists in orthodontics and cleft anomalies, dentists and maxillofacial surgeons proceeded with further evaluation. To prepare the prototype for the procedure of intraoral scanning, extraoral 3D photography, in this case CT imagining, was also performed and a prototype was designed in the 3D program Invivo 6 (Anatomage Inc., San Jose, CA, USA), and manufactured by using a 3D printer and 3D biocompatible resin. The process of design, additive manufacturing and postprocessing is described in detail later.

Application of the intraoral and extraoral fixating device was applied under the supervision of a pediatric ENT specialist and a maxillofacial surgeon. Application of the intraoral device was supervised by proceeding with video-fibro-endoscopy, and the relation to the root of the tongue, structures of the hypopharynx, cleft palate and the position of the spur were evaluated. Modifications were performed during the procedure as it was the first model manufactured for a toddler with some level of self-awareness, prone to have an immediate vomiting reaction to anything inserted into the mouth or present close to the mouth. Multiple samples of the device were printed and modified by shortening the spur and modeling different sizes, shapes, and lengths of the spur, to start with habituation and to continue with the best possible effect on the airways.

During the first trial under observation with the use of video-fibro-endoscopy, the device had an immediate effect on the airways and no vomiting reaction from the toddler. In the next session, the toddler did not cooperate well; there was a vomiting reaction to anything close to her face, so the team decided, after consultation with a speech therapist, on the habituation, starting from the plate with no spur and progressing to a plate with the full length of the spur, as the patient has a severe mental disability without the possibility of coming to realization and understanding. Most importantly the effect on the airways was positive from the very first device, with perfect fit and adaptation, without any negative impact on the mucosa function or swallowing.

### 2.2. Practical Aspects: Patient, Scanning, Virtualisation and 3D Appliance Design

The presented example case of two-year-old girl with diagnoses of different combinations of chromosomal aberrations had no specific genetic diagnosis of isolated Pierre Robin sequence. This case was chosen as a scenario for the demonstration of the utilization of a full-body composite resin appliance design, manufacturing, and application as all main features of PRS were present: micrognathia, cleft and glossoptosis with airway constriction. The exact causes of Pierre Robin syndrome are unknown. Changes in the DNA near the SOX9 gene are the most common genetic cause of isolated cases of Pierre Robin sequence. Pierre Robin sequence is a condition present at birth, in which the infant has a smaller-than-normal lower jaw (micrognathia), a tongue that is placed further back than normal (glossoptosis), and an opening in the roof of the mouth (cleft palate). This combination of features leads to difficulty breathing early in life. Pierre Robin sequence may occur alone (isolated) or be associated with a variety of other signs and symptoms (described as syndromic). In about 20 to 40 percent of cases, the condition occurs alone [33].

The advantage of the presented concept is that the design and manufacturing of the 3D appliance remain completely in the hands of the clinician. The method is feasible even with basic knowledge of 3D computer modeling and operation of 3D printers. Two sources of morphological 3D data are necessary:Intraoral scan (provides highly accurate info about gums and teeth if present)Head CT scan (provides images—slices in shades of grey in DICOM format)

To gain 3D information from CT slices, they need to be segmented first. CT segmentation can be performed in 3D Slicer (www.slicer.org (accessed on 11 September 2022)), a free, open-source and multi-platform software package widely used for medical, biomedical, and related imaging research. The method presented in this paper used software Invivo 6 (Anatomage Inc., San Jose, CA, USA) for CT segmentation with the output in STL file format. The appliance presented in this paper had two parts, connectable with a simple click-in sphere joint:The intraoral part was in contact with the pharynx gums, teeth, tongue, and palateThe extraoral part was in contact with the forehead, base of the nose and cheeks

The appliance to be designed and manufactured in the example case of craniofacial disorder (PRS) was supposed to achieve an effect comparable to the effect of the conventional Tübingen Palatal Plate (TPP) in Figure 3. With the original TPP with wire, the infant also learns to suck milk, so it can be in place basically around the clock, which would not be possible with our new design. For simplification, the new 3D-printed two-part appliance described in this article has not been named in any special way and is referenced as “TPP” as well later in the text. The manufacturing of conventional TPP is a complicated process and typically, the laboratory provides two alternatives with different spur angulation as a rough anticipation of suitable proportions. Old TPP is made from physical intraoral impressions and CT segmentation is also not implemented in the process.

The principle of therapeutical effect in this appliance is the propulsion of the root of the tongue forward and the consequent opening of the collapsed airway in the corresponding portion of the pharynx. The scheme in Figure 4 shows three main parts of the appliance, where the pressure of the oropharyngeal spur to propel the tongue has alveolar anchorage with extraoral support. The 3D-printed appliance is removable, with one extraoral part and one intraoral.

Only the parts in contact with the appliance were relevant from the CT and were exported in STL format and later aligned with the STL output from the intraoral scanning, as is shown in Figure 5a,b.

The correct alignment of the intraoral scan with the CT is very important, as the accurate intraoral scan provides detailed information about the location where the appliance will be placed. Misalignment might lead to an extremely different orientation of the spur from what was intended. This can lead to the patient choking, extreme gag reflex or damage to the mucous membrane of the pharynx. The alignment of the segmented CT and intraoral scan can be performed in various programs, including Meshmixer™ (Autodesk^®^, Inc., San Rafael, CA, USA), as is shown in Figure 6.

The 3D designing of the appliance presented in this paper was performed in a free and practical program, Meshmixer™ (Autodesk^®^, Inc., San Rafael, CA, USA).

The invasiveness of the presented concept lies in the necessity of the initial CT scan, albeit some experts do not consider x-ray examination as an invasive examination method. CT uses ionizing radiation that may cause damage to DNA, and children are at greater risk of carcinogenesis due to their higher tissue radiosensitivity and their longer life expectancy compared to adults. The costs and benefits of the examination for the patient must be clear. In life-threatening situations involving severe craniofacial syndromes, a CBCT or CT scan is performed working from other indications. Figure 7. shows the intention of pediatric otorhinolaryngology department to provide orthodontists with a guide for the approximate ideal length of the spur, visualized on the sagittal slice of the CT scan (Figure 7). The method presented in this paper provides information about 3D patient virtualization. The airway entry with the spur must recognize not only a potential maxillary cant, but also any asymmetries or lateral excursions of the airway where the spur will be oriented. A majority of these can be evaluated without the necessity of CBCT/CT diagnostics. This paper does not define 3D x-ray examination as a necessity. The described method enables 3D design based on segmented CT or CBCT scan.

The second source of 3D data to segmented CT is the intraoral scan. In the presented case, iTero (Align Technology, San Jose, CA, USA) was used. The scan is provided by a dentist under the supervision of an otorhinolaryngology doctor and a maxillofacial surgeon at the hospital, to ensure a safe environment in case of complications.

A complementary extraoral scan was performed —3D photography, acquired by an iPhone 13MAX with a Polycam app, was used to create a surface face scan to provide morphological data of the face surface for the extraoral part of the appliance. This was not a feasible scenario due to the necessity of patient compliance, including head stillness and avoiding changing facial expressions. The surface of the face segmented from CT was sufficiently accurate. The primary significance of the extraoral part is to support anchorage on the alveolar part of the appliance to balance the pressure of the spur being pushed dorsally by the tongue. The secondary significance is its function as a safety feature, preventing the patient from inhaling or swallowing the intraoral part of appliance due to the attachment between the extraoral (EO) and intraoral (IO) parts in the form of a simple spherical click-in joint.

Segmentation of the airway in the area between the nasopharynx and oropharynx is important to plan the spur’s position, especially in craniofacial syndromes with palatal cleft. Figure 8 shows a CT image of the segmented airway showing the complete morphology from the epiglottis up to the nasopharynx.

After the 3D models alignment, the following steps shall be performed in Meshmixer to create a personalized appliance suitable for additive manufacturing:Alveolar part creation (the body)
Region definition (Figure 9.)Joint placement for EO to IO attachment (Figure 10)
Pharyngeal part creation (the spur)
c.Midline and axis identification (Figure 11a,b)d.Spur profile design (Figure 12 and Figure 13)Connecting the body and the spur (Figure 14)Extraoral part creation
e.Region definition (Figure 15)f.Lip bow creation (Figure 16)g.Forehead button and anti-decubital surface creation (Figure 17)



All the following steps (Figure 9, Figure 10, Figure 11, Figure 12, Figure 13, Figure 14, Figure 15 and Figure 16) are performed in Meshmixer™ (Autodesk^®^, Inc., San Rafael, CA, USA), version 3.5.474. Separate visualized steps lead to the creation of the “Alveolar part”, “spur part” and “extraoral part”. The methodology is universal and suitable for any patient. The methodology of sequential steps creating the body complementary to the morphology of the teeth, gums and palate is repeated at the renewal of the appliance due to the patient’s growth. This set of steps is performed to recreate the actualized version of this part separately without needing to edit the remaining parts of the appliance. New morphological data are provided with a new intraoral scan.

### 2.3. Practical Aspects: Material Dental LT Clear V2, Additive Manufacturing and Post Processing

Additive manufacturing was conducted with an SLA printer (Form 2, Formlabs, Somerville, MA, USA) and Dental LT Clear V2 (Formlabs). After printing, the post-processing protocols specified by the manufacturer were followed for each material. For Dental Clear LT, rinsing was performed for 20 min in a 99% IPA isopropanol ultrasonic bath, air drying for 30 min and 20 min UV-curing performed for 60 min at 60 °C (FormCure, Formlabs). Polymers for conventional, subtractive, and additive manufacturing of occlusal devices frequently differ in hardness and flexural properties, but not in wear resistance [34]. The composition of Dental LT Clear V2 material dedicated for SLA 3D printing is shown in Table 1. Mechanical properties are shown in Table 2. Additive resins frequently show reduced surface hardness, flexural strength and flexural modulus compared with injection molded and milled acrylates [34,35].

Dental LT Clear Resin (V2) has been evaluated in accordance with ISO 10993-1:2018, Biological evaluation of medical devices—Part 1: Evaluation and testing within a risk management process, and ISO 7405:2018, Dentistry—Evaluation of biocompatibility of medical devices used in dentistry and passed the requirements for the following biocompatibility risks shown in Table 3.

From practical aspects, the 3D printing process is standard and follows the instructions of the resin manufacturer. Details of this part of appliance manufacturing are available in Appendix A section of this paper. Figure A1 describes the setting of the objects for SLA 3D printing with Dental LT Clear V2 (Formlabs) on 3D printer Form 2.

Figure A2 shows an example of 3D printing failure where extreme thinness of the extraoral part resulted in material buildup drop.

Figure A3 shows views of post-processing steps specified by the manufacturer, including rinsing (for 20 min in a 99% IPA isopropanol ultra-sonic bath), air drying (for 30 min) and UV-curing (60 min at 60 °C).

From a practical clinical point of view, a new appliance can mean a sudden change for a child. The patient’s acclimation to the final appliance can be facilitated by the gradual use of smaller variants of the final appliance. These habituation variants are worn starting from the most comfortable design and gradually approaching the final shape. In patients with cleft formation, even the smallest appliance provides separation of the oral and nasal cavities (Figure 18—bottom center). The arrow in Figure 18 indicates the different lengths of the appliance’s spur, which are intended to facilitate adaptation to the patient (sequence of such habituation appliances shows Figure 18 on the left).

## 3. Results

The result of patient virtualization based on CT and intraoral scan for extraoral appliance design is shown in Figure 19. The final result of appliance design is shown in Figure 20.

Figure 21a shows the intraoral part of the final appliance after additive manufacturing and post-processing, printed from Dental LT Clear V2 (Formlabs, Somerville, MA, USA). Figure 21b shows the whole appliance. Figure 22 shows clinical application on the patient.

## 4. Discussion

Polymers, together with metals and ceramics, have become a widely researched class of materials for applications in AM. Their synthetic versatility and adaptability, as well as the wide range of properties that can be achieved using polymer materials, have rendered polymers the most widely employed class of materials for AM methodologies [36].

As demonstrated by the results, only the biocompatible photopolymer resin Dental LT Clear (V2) can be used to manufacture the appliance. The mechanical properties of the composite resin provide sufficient biomechanical functionality and biocompatibility for the appliance. Pierre Robin Sequence, like many other craniofacial disorders, renders personalized appliances produced in dental labs obsolete very fast due to the rapid growth of infants. Additive manufacturing (AM) permits the fabrication of fully customized objects with a high level of geometrical complexity with reduced fabrication time and cost [37].

Among the various biocompatible materials used in medicine, the material described in this article can be classified in the group of biocompatible materials in dentistry. The spectrum of biocompatible materials is wide and is currently being expanded. From recently reported non-cytotoxic resins developed for use in custom high-resolution 3D printers [38], to new technologies and materials for tissue engineering of a bioartificial pancreas [39] or PCL/SrHA scaffolds [40], or the 3D-printed PLA and PLA-CaCO3 composite splints for the treatment of fractures recently published by Schlégl et al. 2022 [41], to techniques for 3D printing of micro-scale structures [42]. Various medical applications have been very useful during the COVID pandemic, including biocompatible mask holders or special 3D-printed aerosol suction parts [43].

The results presented on the case can inspire a wide community of medical experts to address disorders in their patients with the presented methodology within the presented concept. The advantages of rapid and feasible update of the appliance design to growing morphology as well as affordable AM of multiple clones for each appliance for practical reasons must not be neglected. It has been a long journey from the early adoption of AM in medical application to the first suitable materials now arriving, and polymers, with their variability in mechanical properties, lead the adoption process. The concept presented in this paper explains the methodology in detail and applies it to a real PRS case.

A rapidly increasing demand for medical products based on biomaterials and tissue engineering is now leading to an extensive growth in biomedical research in these fields. A highly interesting class of biomaterials are polymer-based composites, which are already widely used in biomedical applications due to their outstanding physical and mechanical properties [37,44].

“The care of youngest patients with craniofacial anomalies by means of digital workflows offers the opportunity to improve the therapy of these patients and allows safe, reproducible, and precise implementation of requirements. It enables translation from conventional impression methods, which are associated with high effort and diverse risks for this patient clientele. Due to the high potential for personalization, there are more possibilities of care even for complex cases. The successful application and development of such a workflow is the result of the synergetic cooperation of an interdisciplinary team as neonatology, orthodontics, dental technology, and engineering.” [45].

The concept, its accuracy and complexity, creation of 3D virtual models, their fusion, 3D printing with use of biocompatible materials suitable for use in patients will soon become an everyday routine, lowering the invasiveness of the treatment for neonates and children with disabilities and craniofacial malformations.

The limitation of this paper is that it constituted a short-term evaluation of clinical effects and the necessity to evaluate a wide interaction of investigated material with a common patient environment. Another limitation to be considered is the safety features of the material and the device, as the fracture of the device intraorally could lead to inhalation, leading to suffocation or to indigestion. The material used for 3D printing of the appliance in this presented method was Dental LT clear from Formlabs. This material was evaluated in the study of Aretxabaleta et al. [31] in the evaluation of the fracture load of orthodontic appliances for PRS treatment in a digital workflow. It has shown the highest fracture load and best breaking behavior among AM materials.

The data from the facial scan provided sufficient output for the printing of the extraoral part of the device. An intraoral scan provides efficient and detailed 3D images used daily in many orthodontic and dental clinics. The design requires knowledge of the functional anatomy of the face, also in respect to the movements of facial muscles. The design of these appliances might be a semiautomated process supported with AI design algorithms in the future [46].

With the data and the knowledge, combined extraoral and intraoral devices can bring improvement in treatment, its effectiveness, and the comfort of the patient. The role of polymers and composites in medical additive manufacturing will only grow and new materials will arrive with even unexpected properties including enhanced antimicrobial performance or 4D properties that will further widen the possibilities of medical applications [16,47].

The workflow presented in this paper describes universal methods independent of the material. Although the material Dental LT Clear V2 has excellent mechanical properties for such an intraoral appliance, its advertised biocompatible properties should be rigorously evaluated to exclude any potential risks of deploying insufficiently characterized materials for biomedical applications in children. The authors express their concerns despite the material ISO-certification of biocompatibility and commercial marketing for use in dental applications. Unfortunately, the official Safety Data Sheet of the Dental LT Clear V2 manufacturer—Formlabs (Somerville, MA, USA) [48] declares a Bisphenol A dimethacrylate ingredient representing 50–70 of the weight % of this resin.

It should be clear if and how much Bisphenol A is leaching from the polymer matrix over time. These data are not provided by the manufacturer and are crucial for any general recommendation of this material for this use case. The authors encourage the scientific community to engage in such research.

It is well known that, together with Methacrylate Monomer(s), Bisphenol A causes skin irritation and can cause serious eye irritation and allergic skin reaction. More importantly, Bisphenol A (BPA) is a proven endocrine disruptor, albeit a key building block of polycarbonate plastic and a precursor for the manufacturing of monomers of epoxy resins. Due to its hormone-like properties, BPA may bind to estrogen receptors, thereby affecting both body weight and tumorigenesis. BPA may also affect metabolism and cancer progression by interacting with GPR30, and may impair male reproductive function by binding to androgen receptors.

BPA exposure in the general population is via food because of the use of BPA in food packaging and via skin because of contact with thermal paper. Most of the population (91–99%) has detectable levels of BPA-conjugates in their urine. BPA is also present in medical devices including implants, catheters, tubing, and various dental materials [49].

Resin-based dental filling materials are relevant sources of exposure, although according to older studies the amount and potential risks are not clear. Among many endocrine-disrupting chemicals, bisphenols including BPA are currently at the forefront of discussion. The substance was synthesized approximately one hundred years ago. Löfroth et al. 2019 concludes in their literature review that “There is leakage of BPA from some dental materials. Bis-DMA contents might convert to BPA in the oral cavity and that there is a contradiction between in vitro and in vivo studies concerning BPA leakage” [50].

In January 2011, the European Commission prohibited the use of BPA in the manufacture of polycarbonate infant feeding bottles. In February 2018, the EU introduced stricter limits on BPA in food contact fenolmaterials, derived from the temporary tolerable daily intake set by EFSA in 2015. Denmark and Belgium have a ban on BPA in food contact materials for infants and young children; Sweden banned it in coatings and varnishes in food contact materials intended for infants and young children; France banned the chemical in all food contact materials in January 2015.

Bisphenol A (BPA)-based monomers are common in dental resin-based materials. The highest amounts of BPA are released from conventional composites [51]. Dental resins are not inert in the oral environment and may release monomers and other substances such as Bisphenol A (BPA) due to incomplete polymerization and intraoral degradation [52]. Bisphenol A is used in many fields of routine clinical dental practice such as restorative dentistry and orthodontics. Regarding Bisphenol-A exposure from dental materials, the current data conclude that it is below the Tolerable Daily Intake levels, but further evaluation is needed to reveal any possible adverse events caused by low-dose BPA exposure [53].

In the study of Kotyk et al. 2014 quantifiable amounts of leached BPA were observed from a thermoformed orthodontic retainer material (7.63 µg/g of material) and an orthodontic adhesive (2.75 µg/g of material). BPA leaching was only observed within the first 3 days of artificial saliva immersion. This paper presents proof of concept to address PRS complications in infants. The exposure of BPA to their growing and developing bodies may be more hazardous than exposure to adults [54]. On the other hand, the situation in which the device is made is often life-threatening.

In an opinion published in December 2021, the European Food Safety Agency (EFSA) has re-assessed the risks of bisphenol A (BPA) in food, suggesting a drastic reduction in the tolerable daily intake (TDI) compared to its previous assessment in 2015. EFSA is now recommending that the TDI be changed from 4 micrograms per kilogram of body weight per day to 0.04 nanograms per kilogram of body weight per day [55].

Based on European Food Safety Authority (EFSA) evaluations the current recommendation for Tolerable Daily Intake (TDI) for BPA of **0**.04 ng/kg bw/day. If adopted, it represents a very substantial reduction (100,000-fold) compared with the existing TDI.

Bisphenol A is used in many fields of routine clinical dental practice such as restorative dentistry and orthodontics. BPA is permitted for use in food contact materials in the European Union (EU) under Regulation 10/2011/EU, relating to plastic materials and articles intending to come into contact with foodstuffs. In January 2011, the European Commission prohibited the use of BPA in the manufacture of polycarbonate infant feeding bottles (as BPA was observed to leach from a silicone baby bottle nipple [56]). In February 2018, the EU introduced stricter limits on BPA in food contact materials, derived from the temporary tolerable daily intake set by EFSA in 2015.

On 6 April 2022 in Helsinki, European Chemical Agency (ECHA) and the Member States have assessed a group of 148 bisphenols and recommended that more than 30 bisphenols need to be restricted due to their potential hormonal or reprotoxic effects. Many bisphenols are known endocrine disrupters both for human health and the environment. Three bisphenols (bisphenol A, bisphenol B and 2,2-bis(4’-hydroxyphenyl)-4-methylpentane) have already been identified as substances of very high concern. Bisphenols with similar uses and functions were defined as a group, companies can use this information to avoid replacing one bisphenol with another which is dangerous.

Most orthodontic adhesive materials are derived from BPA. The BPA configuration forms a bulky, stiff chain that provides low susceptibility to biodegradation and considerable strength and stiffness in BPA-derived dimethacrylate polymers based on monomers such as bisphenol A glycidyl dimethacrylate (BisGMA), its ethoxylated analog (BisEDMA), bisphenol A dimethacrylate (BisDMA), and urethane-modified BisGMA. Although BPA is not used as a raw material in dental composite resins, it is likely present as an impurity from the chemical synthesis process.

In orthodontics, BPA dimethacrylate derivatives are mostly used for bonding brackets (bonding resins and composite resins as main adhesives) and lingual retainers, whereas BPA-polycarbonates are used for manufacturing plastic brackets. In vitro studies have documented the release of BPA from polycarbonate brackets, orthodontic adhesives, and the composite resins that are frequently used for bonding lingual retainers. For traditional and flowable composite resins used as lingual retainers, BPA release was confirmed in vivo as well, with the highest values in saliva measured immediately after polymerization.

Efforts have been made to replace BPA monomer derivatives with other BPA-free monomers with the goal of achieving the proven stiffness, strength, rigidity, and low biodegradability of BisDMA derivatives. Most alternative approaches included aliphatic comonomers based on triethylene glycol dimethacrylate, urethane dimethacrylate and cycloaliphatic dimethacrylates, all derived from restorative composite resin technology, and suitable filler particles as reinforcing agents [57].

Even from “BPA-free” composites, BPA is often released, albeit in much smaller amounts than from Bis-GMA-containing composites. Despite incubation in methanol, the amounts of BPA detected were well below the applicable limits, indicating that dental composites should not pose a health risk when polymerized appropriately [58]. It has been confirmed that low pH negatively affects the release of BPA. Therefore, frequent exposure to low pH due to consumption of various beverages after sealant treatment may negatively affect the chemical stability of the sealant in the oral cavity [59]. Composite materials may be considered a potential long-term source of BPA and therefore should not be neglected when assessing overall exposure to endocrine disrupting chemicals. [60]. Traces of BPA are present as an impurity in unpolymerized composite materials [61].

Orthodontic polymers and their applications have been instrumental in introducing esthetics, innovation, and practicality to orthodontics. These materials form a large class of components, including resin elements and auxiliaries such as adhesives, polycarbonate brackets and aligners.

The presented concept and methods demonstrate successful design and fabrication using a material with suitable mechanical properties. The main limitation remains the biosafety of biocompatible resins for 3D printing.

## 5. Conclusions

The mechanical properties of Dental LT Clear (V2) are suitable for additive manufacturing of complex orthopedic devices for the treatment of airway narrowing caused by microcrognathia as part of the Pierre Robin sequence.

This proof-of-concept study demonstrates that orthopedic devices manufactured entirely with additive manufacturing can be designed from a 3D combination of intraoral scans aligned with segmented CT or CBCT.

Biomedical applications of photocomposite resins are changing the paradigm of interdisciplinary treatment of craniofacial disorders, including the Pierre Robin sequence, with undeniable benefits from 3D-printed personalized appliances that lend themselves to frequent design updates.

The authors note the potential risks of using biologically under-characterized materials for biomedical applications and emphasize the need for more rigorous evaluation and designation of biocompatible materials. The risks are significantly higher when treating infants than adults.

The concept represents a viable approach to the use of 3D printing in the manufacture of personal medical devices and is independent of the composite material used. The material used in the methods is based on bisphenol A dimethacrylate with unknown resistance to its leaching. The outlook for dental materials based on bisphenol derivatives is not optimistic, and the authors anticipate a legislative ban on broad groups of bisphenols in the EU in the near future.

## Figures and Tables

**Figure 1 polymers-14-03858-f001:**
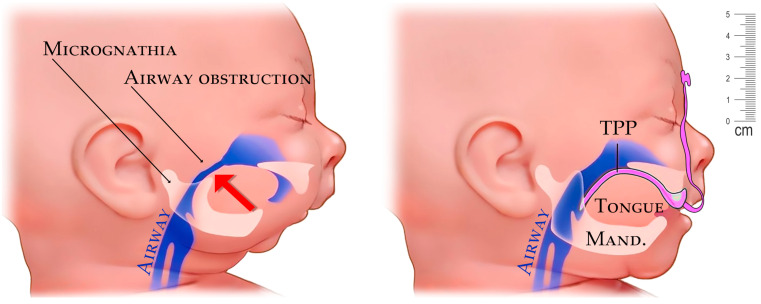
Example of airway stenosis (constriction) in a newborn caused from the pressure of the tongue from undersized mandible in micrognathia (**left**). Improvement of airway with placement of a personalized appliance—Tuebingen Palatal Plate (TPP) with extraoral part (**right**).

**Figure 2 polymers-14-03858-f002:**
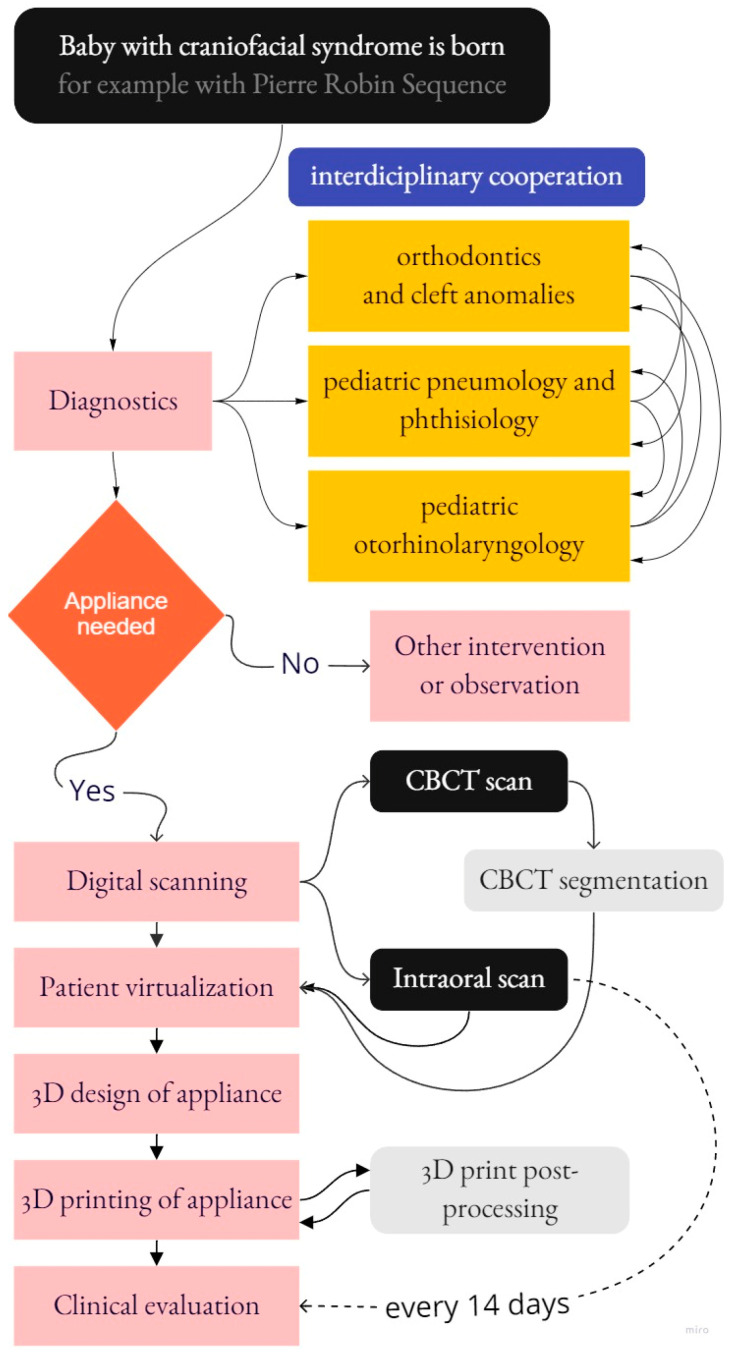
The complex scheme of processes covered by this paper.

**Figure 3 polymers-14-03858-f003:**
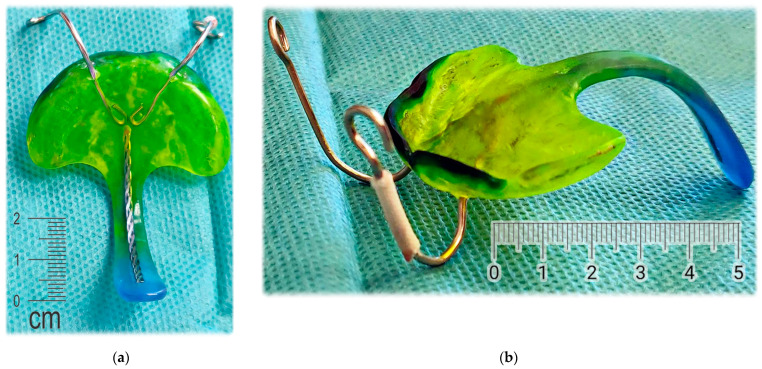
Conventional design of Tübingen Palatal Plate (TPP) made by laboratory technician with metallic hooks for extraoral pull and safety metallic wire in the body of oropharyngeal spur: (**a**) Lingual view; (**b**) Palato-lateral view. Scale is in cm.

**Figure 4 polymers-14-03858-f004:**
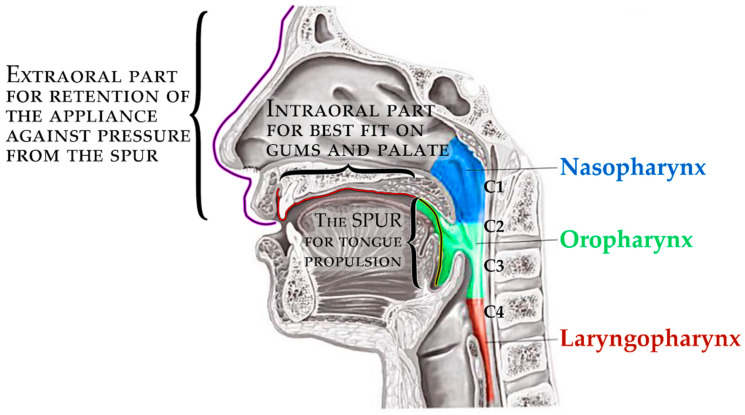
The complex scheme of processes covered by this paper.

**Figure 5 polymers-14-03858-f005:**
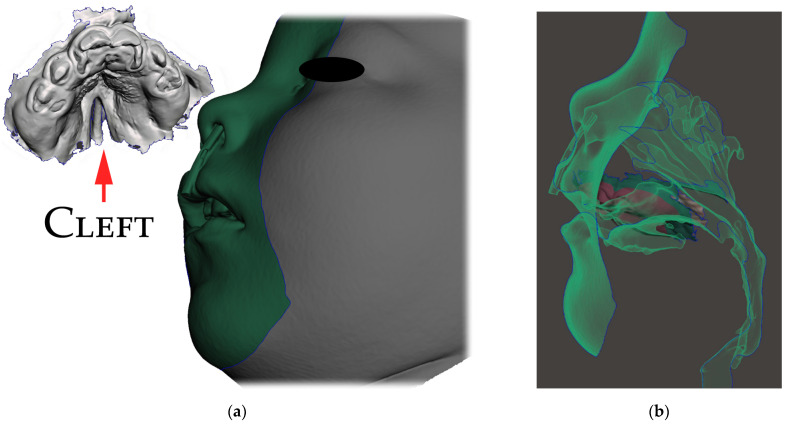
Segmented CT and intraoral scan: (**a**) surface of the face, intraoral cavity and relevant portions of the pharynx were segmented from CT to be aligned with the highly accurate intraoral scan of the upper alveoli and palate with the cleft on the left; (**b**) aligned intraoral scan with the rest of the segmented CT.

**Figure 6 polymers-14-03858-f006:**
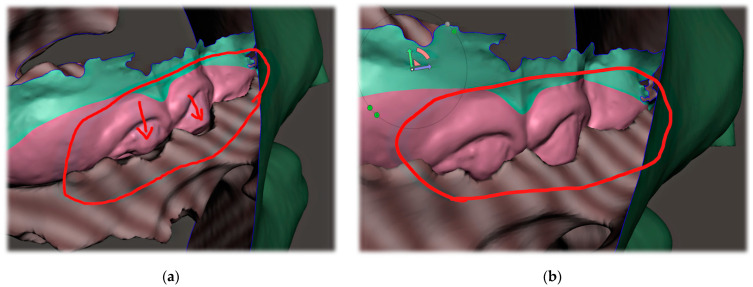
The alignment of the intraoral scan with segmented CT: (**a**) automatized alignment error found; (**b**) alignment of intraoral scan of the upper arch was corrected in Meshmixer™ (Autodesk^®^, Inc., San Rafael, CA, USA).

**Figure 7 polymers-14-03858-f007:**
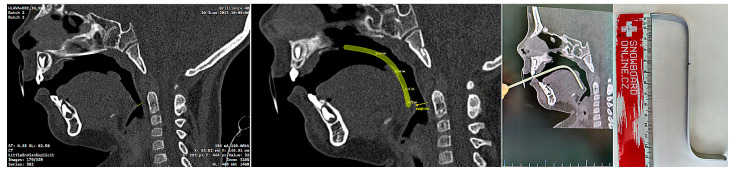
Example of a legacy CT utilization by a pediatric otorhinolaryngology department in an effort to provide instructions for orthodontists and dental laboratories for the approximate ideal length and angulation of the spur. CT/CBCT examination of the patients is not necessary to identify the morphology and relationships of the soft tissues, although it can be successfully utilized for the 3D design of the appliance.

**Figure 8 polymers-14-03858-f008:**
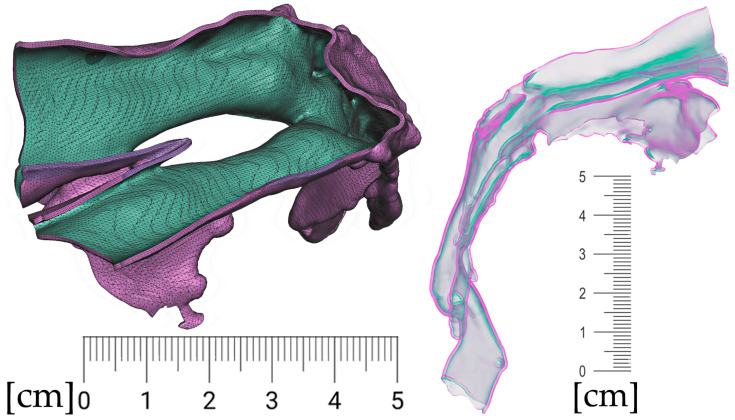
Segmentation of the airway from CT scan in the area between nasopharynx and oropharynx; knowledge of the morphology is important for the design of the spur shape, especially in craniofacial syndromes with palatal cleft.

**Figure 9 polymers-14-03858-f009:**
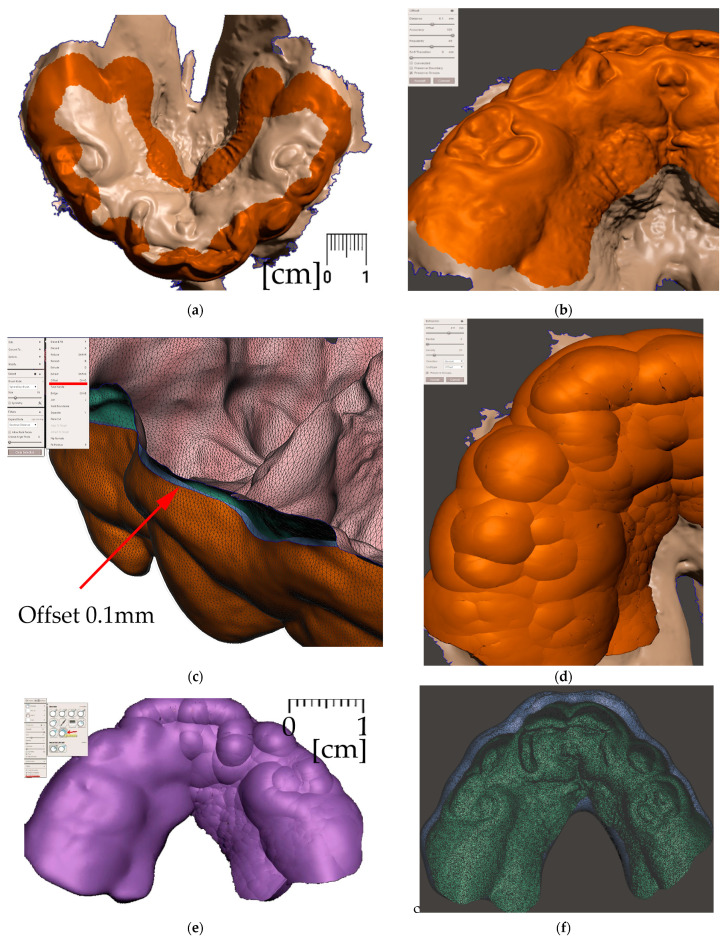
Steps to create the Alveolar part (the body) of the appliance: (**a**) selection tool -boundaries definition. (**b**) Select tool, -> optimize boundaries—keystrokes “O”, “B”; (**c**) Edit—Offset tool by 0,1 mm creating a separated copy of the morphology; (**d**) Edit—Extrusion keystroke “D” direction “normal”; (**e**) automatic: Select group and keystroke “Ctrl+F” smooth or Manual: Sculpt—Surface brush—“Robust smooth”—filters “restrict to group”; (**f**) Final 3D Alveolar body of appliance.

**Figure 10 polymers-14-03858-f010:**
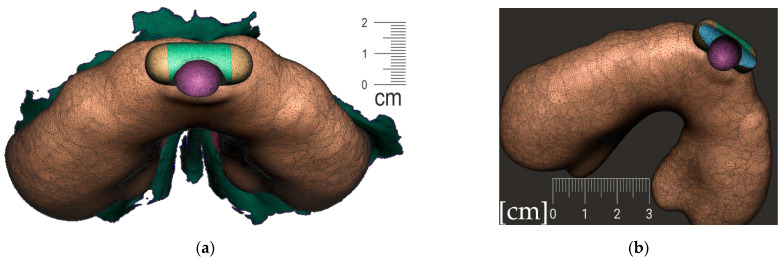
A Joint placement for EO to IO attachment: (**a**) frontal view with intraoral situation including the view of the cleft; (**b**) lateral view of the spherical joint attachment.

**Figure 11 polymers-14-03858-f011:**
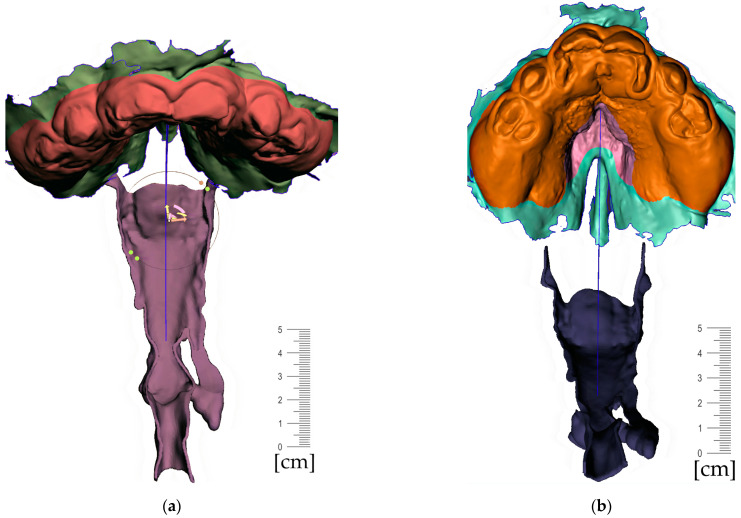
Pharyngeal part creation—the spur. Midline and axis identification: (**a**) frontal view—the midline of the future spur shall be centered; (**b**) palatal view of the midline of future spur shall be connecting the future appliance’s alveolar body symmetrically in the middle of the alveolar arch.

**Figure 12 polymers-14-03858-f012:**
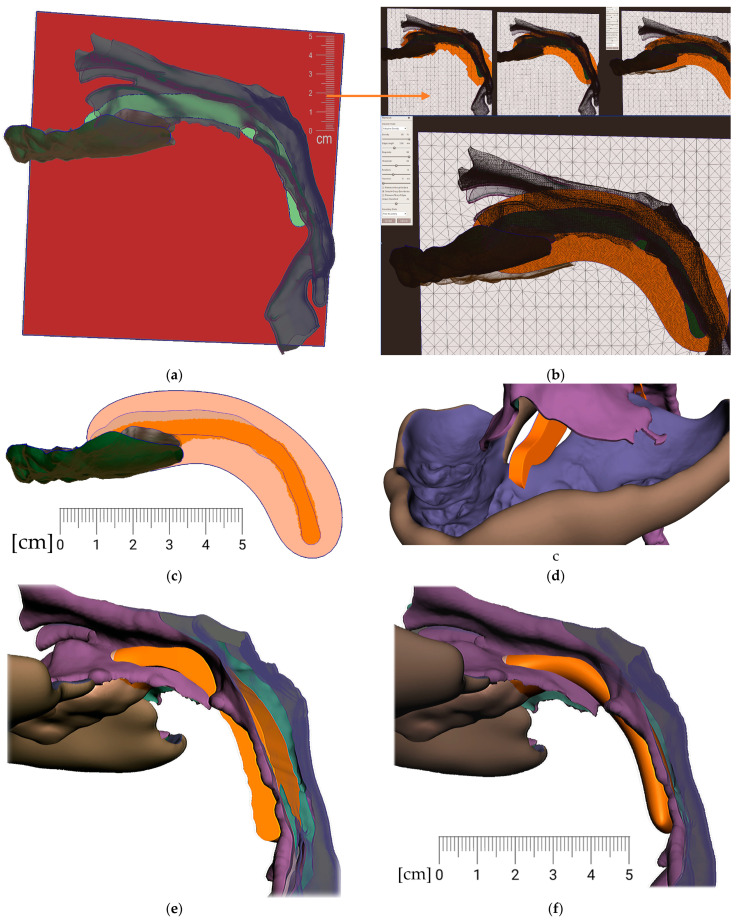
Pharyngeal part creation (the spur)—Spur profile design—Region definition: (**a**) import a Mesh primitive—2D plane; (**b**) select tool define region -> optimize boundaries—keystrokes “O”, “B” and Re-mesh to high density key strokes “R”; (**c**) select and delete irrelevant part of the plane; (**d**) Edit—Extrusion keystroke “D”; (**e**) extrude to desired width; (**f**) select all and keystroke “Ctrl+F” smooth.

**Figure 13 polymers-14-03858-f013:**
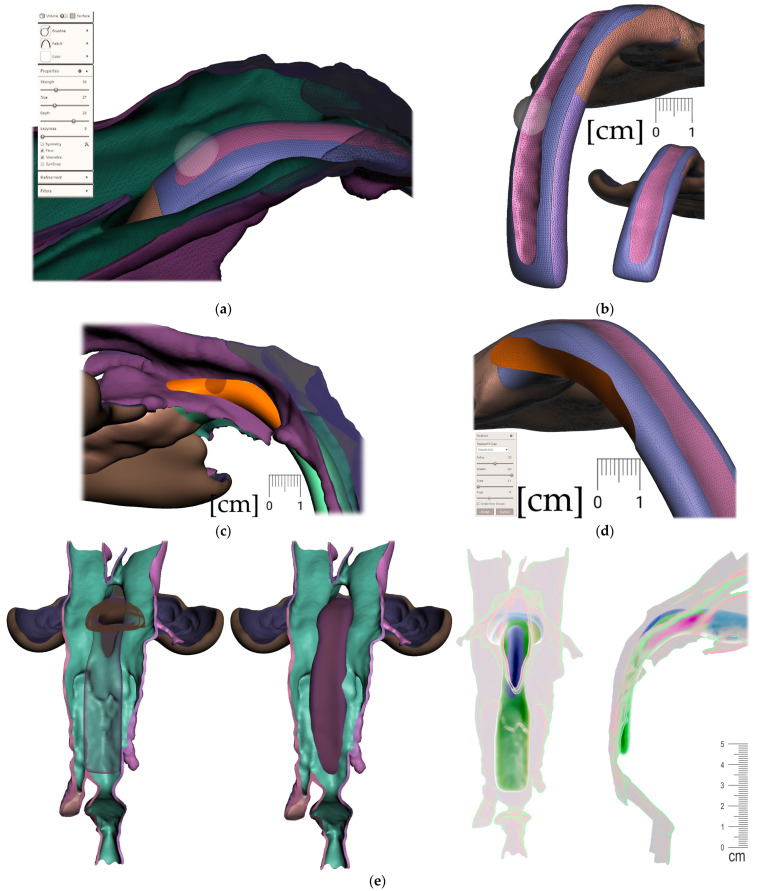
Pharyngeal part creation (the spur)—Spur profile anatomical adaptation: (**a**) select Sculpt tool and Volume Brush “Drag” define shape adaptations of the spur; (**b**) longitudinal groove impression increases resilience of the spur; (**c**) select and optimize selection– keystroke “O”, and smooth boundaries “B”; (**d**) with selected area Edit -> “Erase and fill” Keystroke “F” to reduce or bulge defined areas; (**e**) posterior and lateral views of spur placement in the pharynx with planned propulsion of the soft tissues in this area. Use Shaders menu to set the spur transparency to evaluate anatomical placement and distance from the epiglottis.

**Figure 14 polymers-14-03858-f014:**
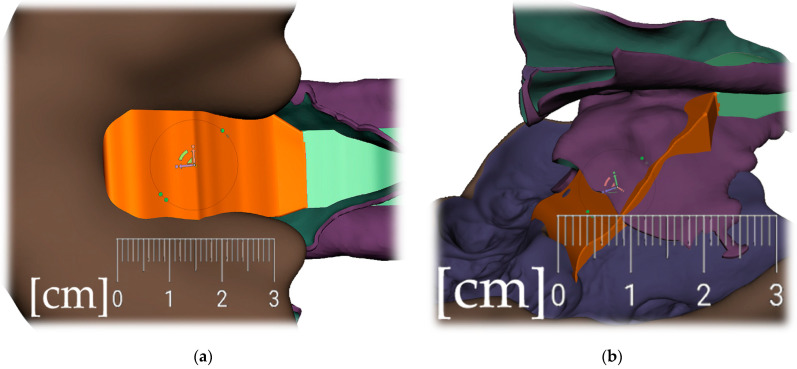
Connecting the body and the spur: (**a**) utilization of Select -> Deform -> Transform tool keystroke “T” to extend the palatal end of the spur to desired width and length; (**b**) transform tool will allow change of axis orientation of the selected part. The remaining mesh differences shall be manually changed via Sculpt -> Brushes -> Robust Brush and final merge of both parts by means of Boolean union after selection of both objects—the Spur part and the Alveolar part of the appliance.

**Figure 15 polymers-14-03858-f015:**
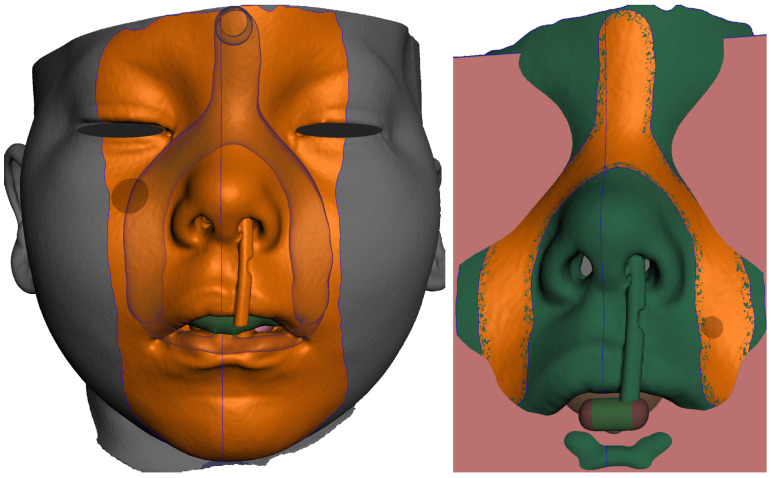
In extraoral part creation—in region definition, the only relevant part of the segmented facial surface from the CT/CBCT is used. Import of 2D plane novelized on the eye pupils can be utilized as a guide for better appliance symmetry. Select tool, -> optimize boundaries—keystrokes “O”, “B” and offset by 0.1 mm from the face surface. Export surface as separate object keystrokes “Y”.

**Figure 16 polymers-14-03858-f016:**
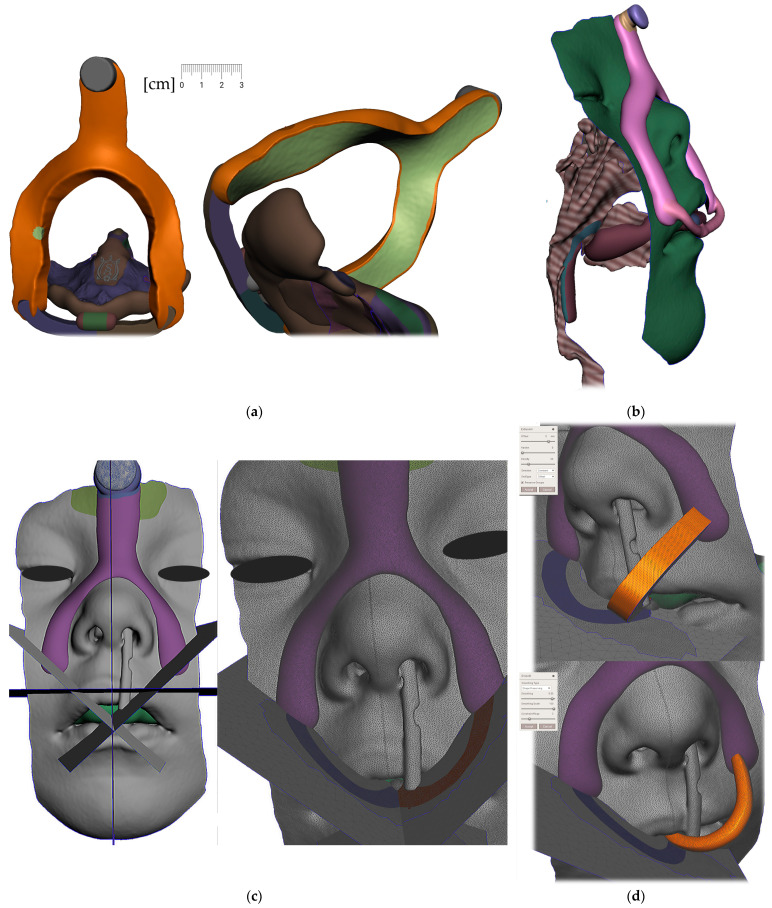
Extraoral part creation—Lip bow creation: (**a**) extrusion of offset of facial surface representing the planned contact with the face followed by smoothing “Ctrl+F” and subsequent extension of the selection keystroke “>” and finished by final smoothing resulting in rounded borders of the extraoral part; (**b**) lateral vied of the variant extraoral part of the appliance; (**c**) Import of mesh 2D plane primitive in 45 ° positioning to allow definition of the labial bow contours; (**d**) extrusion of the contours is followed by smoothing ending in spherical connector to spherical socket.

**Figure 17 polymers-14-03858-f017:**
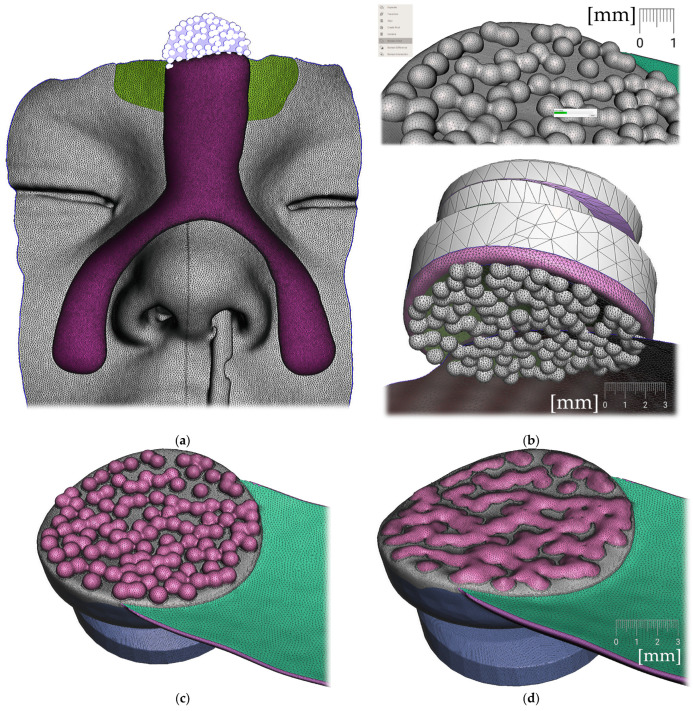
Extraoral part creation —Forehead button and anti-decubital surface creation: (**a**) definition of the pattern of spheres to achieve anti-decubital surface at the level of skin; (**b**) selected area -> Edit -> Make Pattern -> Tiled Spheres as pattern type, Clip to surface; (**c**) select both objects and perform Boolean Union followed by Smoothing; (**d**) final skin-facing smoothed anti-decubital surface of the button for extraoral elastic cap attachment.

**Figure 18 polymers-14-03858-f018:**
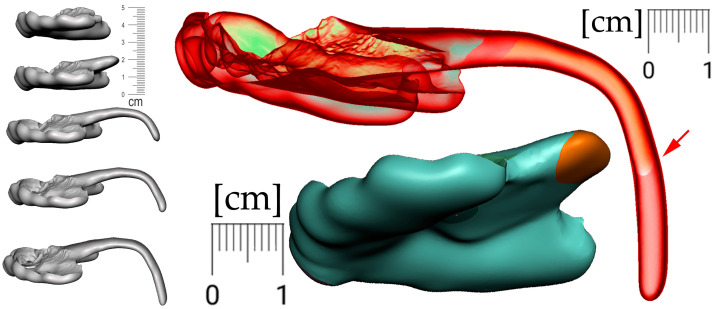
For better patient accommodation, the final appliance can be fragmented into a few smaller variants that are worn from the most comfortable to the final shape. In patients with cleft, even the most reduced appliance provides separation of the oral and nasal cavities (bottom center). The arrow points to appliance different lengths of the spur that help to acclimatize the patient to the device.

**Figure 19 polymers-14-03858-f019:**
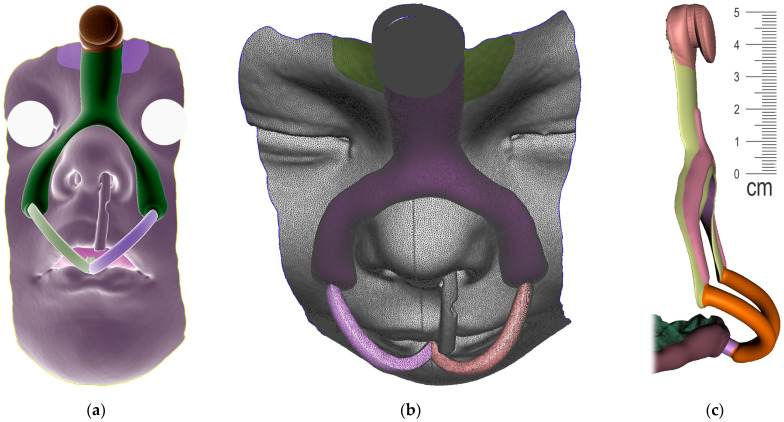
Result of Extraoral part of design based on CT and intraoral scan: (**a**) frontal view of extraoral part of appliance with button on the forehead intended for elastic cap providing retention, complementary nasal part distributing the pressure on wider surface and labial bows ended with spherical joint connecting to Alveolar part of appliance; (**b**) view of composite bows leaving the surface part circumventing the highly movable part of the lips; (**c**) lateral view of the final 3D personalized appliance suitable for 3D printing from Dental LT Clear V2 (Formlabs, Somerville, MA, USA).

**Figure 20 polymers-14-03858-f020:**
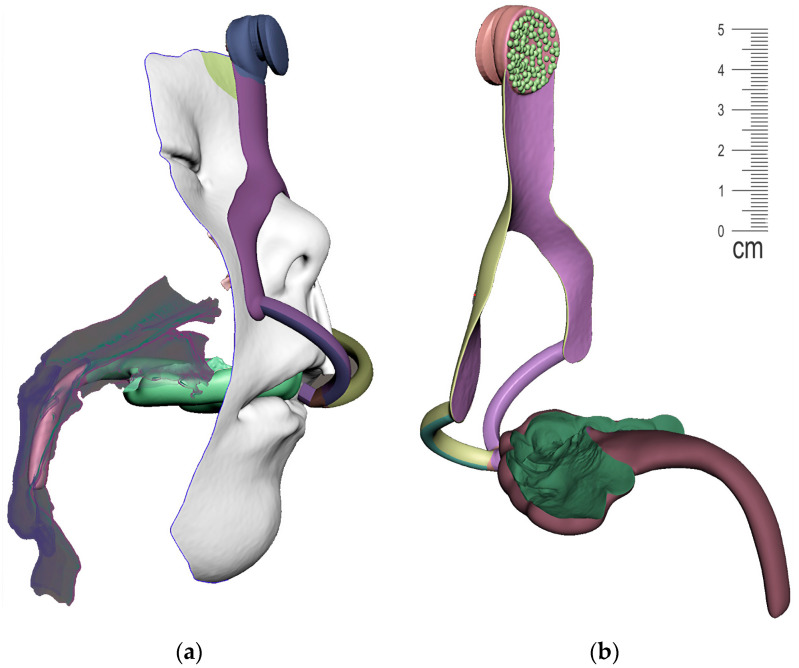
View of the final appearance of the appliance: (**a**) with patient; (**b**) without patient.

**Figure 21 polymers-14-03858-f021:**
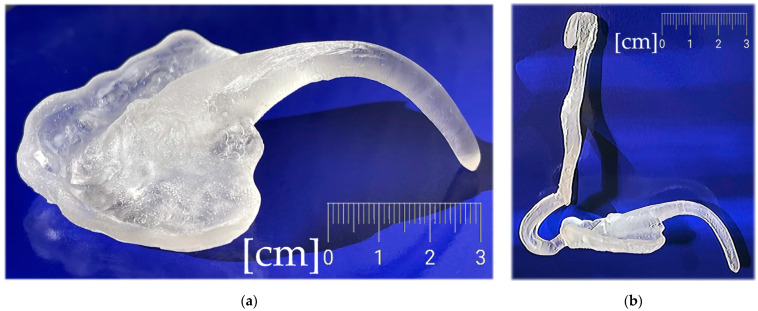
The final appliance after additive manufacturing and post-processing, printed from Dental LT Clear V2 (Formlabs, Somerville, MA, USA): (**a**) the intraoral part; (**b**) complete.

**Figure 22 polymers-14-03858-f022:**
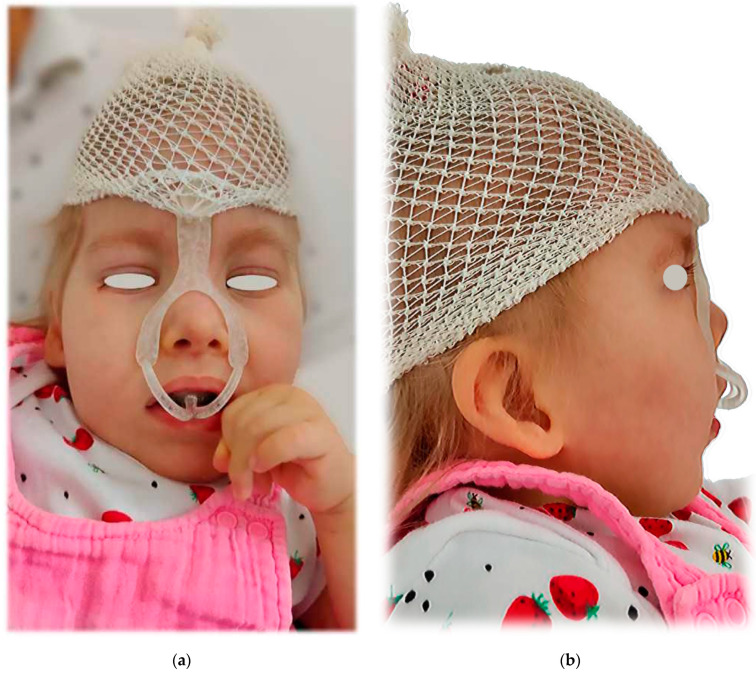
The clinical application on the patient: (**a**) frontal view; (**b**) lateral view.

**Table 1 polymers-14-03858-t001:** Composition of the investigated material.

Composition of Dental LT Clear V2
Bisphenol A Dimethacrylate 50–70%
Methacrylate monomer ➝ 7–10%
Photoinitiator ➝ <2%
Urethane Dimethacrylate ➝ 25–45%

**Table 2 polymers-14-03858-t002:** Mechanical properties of the investigated material Dental LT Clear V2.

Mechanical Properties ^1^	Post-Cured ^2^	Method
Elongation	12%	ASTM D638-10 (Type IV)
Flexural Strength at 5% Strain	84 MPa	ASTM D790-15 (Method B)
Flexural Modulus	2300 MPa	ASTM D790-15 (Method B)
Hardness Shore D	78 D	ASTM D2240-15 (Type D)

^1^ Data were measured on post-cured samples printed on a Form 3B printer with 100 µm Dental LT Clear Resin (V2) settings, washed in a Form Wash for 20 min in 99% Isopropyl Alcohol, and post-cured at 60 °C for 60 min in a Form Cure. ^2^ Data were measured on post-cured samples printed on a Form 3B printer with 100 µm Dental LT Clear Resin (V2) settings, washed in a Form Wash for 20 min in 99% Isopropyl Alcohol, and post-cured at 60 °C for 60 min in a Form Cure.

**Table 3 polymers-14-03858-t003:** Evaluation of biocompatibility of the investigated material Dental LT Clear V2.

ISO Standard	Description
EN ISO 10993-5:2009	Not cytotoxic
ISO 10993-10:2010/(R)2014	Not an irritant
ISO 10993-10:2010/(R)2014	Not a sensitizer
ISO 10993-3:2014	Not genotoxic
ISO 10993-11:2017	Not toxic

Dental LT Clear Resin (V2) was evaluated at NAMSA World Headquarters, OH, USA.

## Data Availability

Not applicable.

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
