# Peer review of "Pierre Robin Sequence and 3D Printed Personalized Composite Appliances in Interdisciplinary Approach"

_polymers, 2022, doi:10.3390/polym14183858_

Round 1

Reviewer 1 Report

1. In the introduction, the authors described different medical conditions with text. However, for persons with no medical background, it is very difficult for them follow the text or visualize the illness. Suggest adding figures for better illustration where applicable.

2. As the medical appliance will be used to take load when it is in use, the creep property of the material becomes an important parameter to investigate. Have the authors conducted any test to study the creep property of the material?

3. According to the SDS of the material, the cured material is biocompatible, how about the biocompatibility of the uncured resin? will it be toxic? how do you ensure that all the resin are completely cured or washed away. I believe there could be certain geometery that the UV light is difficult to reach and the IPA bath may not be able to clean effectively. How do you deal with this kind of situation? Have you evaluated what kind of geometry can result in bad curing and cleaning? suggest discussing in the manuscript.

4. Suggest adding a scale bar to all the figures.

5. Since this manuscript discussed biocompatible materials for 3D printing, suggest discussing all the different biocompatible materials for 3D printing such as hydrogel, etc. Below are some articles that you may consider citing.

a. Warr, C., Valdoz, J. C., Bickham, B. P., Knight, C. J., Franks, N. A., Chartrand, N., ... & Cook, A. D. (2020). Biocompatible PEGDA resin for 3D printing. ACS applied bio materials3(4), 2239-2244.

b. Bell, A., Kofron, M., & Nistor, V. (2015). Multiphoton crosslinking for biocompatible 3D printing of type I collagen. Biofabrication7(3), 035007.

c. Soetedjo, A. A. P., Lee, J. M., Lau, H. H., Goh, G. L., An, J., Koh, Y., ... & Teo, A. K. K. (2021). Tissue engineering and 3D printing of bioartificial pancreas for regenerative medicine in diabetes. Trends in Endocrinology & Metabolism32(8), 609-622.

d. Liu, D., Nie, W., Li, D., Wang, W., Zheng, L., Zhang, J., ... & He, C. (2019). 3D printed PCL/SrHA scaffold for enhanced bone regeneration. Chemical Engineering Journal362, 269-279.

Author Response

Comments and Suggestions for Authors

  1. In the introduction, the authors described different medical conditions with text. However, for persons with no medical background, it is very difficult for them follow the text or visualize the illness. Suggest adding figures for better illustration where applicable.

Dear colleague, thank you for your good point. We have created a special Figure 1, explaining the mentioned clinical condition of micrognathia with and without TPP (lines 53-58).

  1. As the medical appliance will be used to take load when it is in use, the creep property of the material becomes an important parameter to investigate. Have the authors conducted any test to study the creep property of the material?

Thank you for your interesting consideration. We are aware of tendencies of solid materials to move slowly or deform permanently under the influence of persistent mechanical stresses. In regard to our considerations, there is currently no data available in regard to the creep property of the Dental LT clear material and would require further investigation. However, as deformation can occur as a result of long-term exposure to high levels of stress, we have considered the following:

1. The Dental LT clear is certified as long-term material suitable of occlusal splints in adults, where occlusal forces are significantly higher than forces of the tongue the TPP shall handle.
2. As well as these splints have an accurate fit on teeth and gums and in a case of high creep property in the material this would result in deterioration of the fit which has not been observed in any of the publications.
3. Thirdly, the presented proof-of-concept recommends rapid changes of the appliance in the matter of days, where any hypothetical long-term deformation effect has no time to occur.

  1. According to the SDS of the material, the cured material is biocompatible, how about the biocompatibility of the uncured resin? will it be toxic? how do you ensure that all the resin are completely cured or washed away. I believe there could be certain geometery that the UV light is difficult to reach and the IPA bath may not be able to clean effectively. How do you deal with this kind of situation? Have you evaluated what kind of geometry can result in bad curing and cleaning? suggest discussing in the manuscript.

Thank you for another interesting point. You are correct that non-processed resin is toxic and there are strict rules to follow for its handling. There is also a strict prescription how to finish the print with post-processing rules https://dental.formlabs.com/indications/splints-and-occlusal-guards/guide/
The ISO certification is a guaranty that when all steps followed there is no option of remaining unpolymerized resin to persist on the print or in its folds. However, to answer your question practically, with hypothetical situation of a complex multifold object a prescribed 15-minute sonic IPA bath in a possible overkill. The time shall not be longer as the IPA can start dissolving the object. Place the printed parts in a Form Wash filled with isopropyl alcohol (IPA, ≥99%) and wash them for 15 minutes. The operator is responsible for ensuring the parts are fully submerged in IPA when washing. Even in hypothetical situation that by an operator error some unpolymerized resin remains on the object the Post Curing with heat and UV light will hit every part of the object during the prescribed 60 minutes. Special Form Cure provides a cyclic 60°C waves in 60 minutes where it completely cools down between the cure cycles. Not only heat but also UV light goes through the object as it is transparent. And finally, the resulting object is polished and finished by a human as the final stage of control.

  1. Suggest adding a scale bar to all the figures.

We have added scales in SI units to all relevant Figures.

  1. Since this manuscript discussed biocompatible materials for 3D printing, suggest discussing all the different biocompatible materials for 3D printing such as hydrogel, etc. Below are some articles that you may consider citing.
  2. Warr, C., Valdoz, J. C., Bickham, B. P., Knight, C. J., Franks, N. A., Chartrand, N., ... & Cook, A. D. (2020). Biocompatible PEGDA resin for 3D printing. ACS applied bio materials3(4), 2239-2244.
  3. Bell, A., Kofron, M., & Nistor, V. (2015). Multiphoton crosslinking for biocompatible 3D printing of type I collagen. Biofabrication7(3), 035007.
  4. Soetedjo, A. A. P., Lee, J. M., Lau, H. H., Goh, G. L., An, J., Koh, Y., ... & Teo, A. K. K. (2021). Tissue engineering and 3D printing of bioartificial pancreas for regenerative medicine in diabetes. Trends in Endocrinology & Metabolism32(8), 609-622.
  5. Liu, D., Nie, W., Li, D., Wang, W., Zheng, L., Zhang, J., ... & He, C. (2019). 3D printed PCL/SrHA scaffold for enhanced bone regeneration. Chemical Engineering Journal362, 269-279.

We have followed your suggestion and created a paragraph in the Discussion chapter discussing different biocompatible materials for 3D printing as suggested. All references were cited in the text.

Thank you for your time and effort improving this paper.

Reviewer 2 Report

The research article with the title: Concept and Method of Biocompatible Photopolymer Resin 2 Based Additive Manufacturing of Appliances in the Interdisciplinary Management of Pierre Robin Sequence is interesting. Ideas and concept of this work are good. The presentation and the conduction is not suitable. A major revision of this work is mandatory. The following issues should be addressed:

1.       The actual manuscript is highly overloaded!!!!! 23 Figures and 3 tables in one manuscript! It is very difficult to find a logic and to follow the structure of this work. The conclusion is: Rewrite and resubmit! Restructure the main manuscript to be more reader friendly. Use a supporting manuscript if it is possible and allowed from the journal, because there are some figures, which are not necessary to be shown in the main manuscript. For example, figure 16 to 18. This is only makes the manuscript more bulky and disturbs the reading flow. 

2.       Check the manuscript carefully, sometimes space signs are missing.

3.       Figure 2: a) and b) are missing.

4.       In all figures which are not showing a scheme (figure 3 for example) the scale bars are missing. Please add the missing scale bars. For example, in figure 2, the readers should know what are the real dimensions of the TPP. 

5.       In this study Bisphenol A Dimethacrylate 50–70% is used. Also if it is a derivate of the highly cangerogen Bisphenol A, it should be clear how much Bisphenol A itself is released from the polymer matrix over time (minimum for one month). There are some strict regulations by the EU (the authors are from the EU) for this compound. It should be clear whether the limits of these regulations are also observed here and it should be concluded not only for adult patient here. Especially, this is of interest for 2-year-old infants where risk of the exposition to Bisphenol A by this specific additive manufactured polymeric product should be taken into account.

6.       The Conclusion chapter is far too short and therefore no conclusion suitable for a scientific paper. Deal with the results obtained in an appropriate way and give an outlook how to deal with the gained results.

Author Response

Comments and Suggestions for Authors

The research article with the title: Concept and Method of Biocompatible Photopolymer Resin 2 Based Additive Manufacturing of Appliances in the Interdisciplinary Management of Pierre Robin Sequence is interesting. Ideas and concept of this work are good. The presentation and the conduction is not suitable. A major revision of this work is mandatory. The following issues should be addressed:

  1. The actual manuscript is highly overloaded!!!!! 23 Figures and 3 tables in one manuscript! It is very difficult to find a logic and to follow the structure of this work. The conclusion is: Rewrite and resubmit! Restructure the main manuscript to be more reader friendly. Use a supporting manuscript if it is possible and allowed from the journal, because there are some figures, which are not necessary to be shown in the main manuscript. For example, figure 16 to 18. This is only makes the manuscript more bulky and disturbs the reading flow. 

Dear colleague, thank you for your feedback. We are aware the paper is complex and difficult to read for non-medical fields. We have improved the Introduction and readability of this paper according your suggestions. We have performed major update on the structure of presentation. As the response on your first point and request of another reviewer we have created a special Figure 1, explaining the mentioned clinical condition of micrognathia with and without TPP (lines 53-58). We hope this will make the paper more reader-friendly even from other non-medical fields. As you have recommended, we have also removed three figures 16, 17 and 18 from the text for improvement of the text readability. These were originally included for medical professionals reading this paper to be able to comprehend the whole text, albeit for technical profession these three pictures could be trivial. For reproducibility and better comprehension of dentists and staff of dental labs, we have preserved them as supplementary in the Appendix of the paper.

  1. Check the manuscript carefully, sometimes space signs are missing.

      Thank you for your observation, we are aware of this error. We have now re-checked the whole paper. We are experiencing this issue which is caused by Mendeley plugin bug/incompatibility. For unknown reasons this occurs during saving the document when two different collaborators edit the document. We will carefully revise this issue in the final syntax check. Thank you

  1. Figure 2: a) and b) are missing.

We have split the image, added the SI unit scale and add A/B description.

  1. In all figures which are not showing a scheme (figure 3 for example) the scale bars are missing. Please add the missing scale bars. For example, in figure 2, the readers should know what are the real dimensions of the TPP. 

We have added scales in SI units to all relevant Figures which were showing unfamiliar objects. Figures showing face, teeth or hand were not augmented with a scale as reader is aware of proportions. We have edited Figures 1, 3, 7, 9, 10, 11, 12, 13, 15, 16, 17. 18c, 19b and 20 a, b.

Original figures 16, 17 and 18 were removed to Appendix and referenced in the text – lines 652-657.

The Figure 17 reference in the text has been rewritten. (Lines 661-667)

  1. In this study Bisphenol A Dimethacrylate 50–70% is used. Also if it is a derivate of the highly cangerogen Bisphenol A, it should be clear how much Bisphenol A itself is released from the polymer matrix over time (minimum for one month). There are some strict regulations by the EU (the authors are from the EU) for this compound. It should be clear whether the limits of these regulations are also observed here and it should be concluded not only for adult patient here. Especially, this is of interest for 2-year-old infants where risk of the exposition to Bisphenol A by this specific additive manufactured polymeric product should be taken into account.

      Thank you for pointing this out. We have elaborated on aspects of this topic with recent scientific findings and murky perspectives of bisphenols for medical applications. We have put these findings in the context of EU legislative and incoming legislative changes. We have emphasized the aspect of age and sensitivity of infants on potential risks that would result from BPA leaching. For evaluation of BPA leaching a new study would be necessary as there is no publication on this matter regarding this particular material. We have taken into account also the wider context of dental resins linked with BPA-derived dimethacrylate polymers. Discussion has been extended for these objectives on lines 803-923.

  1. The Conclusion chapter is far too short and therefore no conclusion suitable for a scientific paper. Deal with the results obtained in an appropriate way and give an outlook how to deal with the gained results.

Our effort was to make the Conclusion chapter as brief and clear as possible. We prefer not to extend it significantly more. Upon your suggestion we have rewritten this chapter and adapted the Abstract as well accordingly. We have added the Outlook as suggested.

Thank you for your time, good points and effort to revise and improve this complex paper.

Round 2

Reviewer 2 Report

The authors of the manuscript entitled Concept and Method of Biocompatible Photopolymer Resin 2 Based Additive Manufacturing of Appliances in the Interdisciplinary Management of Pierre Robin Sequence addressed the most of my comments appropriate. After revision, the quality of the manuscript raised. Still the manuscript is overloaded in my opinion and my concerns about the release of Bisphenol A from the polymer matrix is not investigated but discussed in the main text and the conclusion. In order not to overloaded the manuscript even more, the release study from this material should be encouraged to the authors as a further study to be carried out, as it is important in terms of safety.